# The *Wolbachia* mobilome in *Culex pipiens* includes a putative plasmid

Julie Reveillaud[1], Sarah R. Bordenstein[2], Corinne Cruaud[3], Alon Shaiber[4,5], Özcan C. Esen[5], Mylène Weill[6], Patrick Makoundou[6], Karen Lolans[5], Andrea R. Watson[5], Ignace Rakotoarivony[1], Seth R. Bordenstein [2,7,8] & A. Murat Eren [4,5,9]

*Wolbachia* is a genus of obligate intracellular bacteria found in nematodes and arthropods worldwide, including insect vectors that transmit dengue, West Nile, and Zika viruses. *Wolbachia*'s unique ability to alter host reproductive behavior through its temperate bacteriophage WO has enabled the development of new vector control strategies. However, our understanding of *Wolbachia*'s mobilome beyond its bacteriophages is incomplete. Here, we reconstruct near-complete *Wolbachia* genomes from individual ovary metagenomes of four wild *Culex pipiens* mosquitoes captured in France. In addition to viral genes missing from the *Wolbachia* reference genome, we identify a putative plasmid (pWCP), consisting of a 9.23-kbp circular element with 14 genes. We validate its presence in additional *Culex pipiens* mosquitoes using PCR, long-read sequencing, and screening of existing metagenomes. The discovery of this previously unrecognized extrachromosomal element opens additional possibilities for genetic manipulation of *Wolbachia*.

[1] ASTRE, INRA, CIRAD, University of Montpellier, Montpellier 34398, France. [2] Department of Biological Sciences, Vanderbilt University, Nashville 37235 TN, USA. [3] Commissariat à l'Energie Atomique et aux Energies Alternatives (CEA), Institut de Biologie François Jacob, Genoscope, Evry 91057, France. [4] Graduate Program in the Biophysical Sciences, University of Chicago, Chicago, IL 60637, USA. [5] Department of Medicine, University of Chicago, Chicago 60637 IL, USA. [6] Institut des Sciences de l'Evolution de Montpellier (ISEM), UMR CNRS-IRD-EPHE-Université de Montpellier, Montpellier 34095, France. [7] Department of Pathology, Microbiology, and Immunology, Vanderbilt University, Nashville 37235 TN, USA. [8] Vanderbilt Institute for Infection, Immunology, and Inflammation, Vanderbilt University, Nashville 37235 TN, USA. [9] Josephine Bay Paul Center for Comparative Molecular Biology and Evolution, Marine Biological Laboratory, Woods Hole 02543 MA, USA. These authors contributed equally: Julie Reveillaud, Sarah R. Bordenstein. Correspondence and requests for materials should be addressed to J.R. (email: reveillaud.j@gmail.com) or to A.M.E. (email: a.murat.eren@gmail.com)

Mosquitoes are major vectors of disease-causing pathogens worldwide including viruses such as dengue, West Nile, Chikungunya, Zika, and yellow fever[1–3]. In the absence of effective vaccines and the off-target effects of insecticides used to control mosquitoes, novel vector biocontrol efforts are the focus of intense study[4]. Over the past decade, the widely distributed endosymbiotic alphaproteobacteria Wolbachia has gained attention as a promising mosquito-control strategy. The basic reasons for its heightened attention is that Wolbachia can cause reproductive parasitism whereby the intracellular bacteria in the reproductive tissues can alter sexual reproduction to enhance its maternal spread through host populations at the expense of host fitness[5–7]. Second and notably, native Wolbachia reduce RNA virus replication in the fruit fly Drosophila[8–10], and Wolbachia transinfections into mosquito species (i.e., Aedes aegypti, Ae. albopictus, Ae. polynesiensis, and Anopheles stephensi) similarly result in mosquito lines refractory to various types of pathogenic RNA viruses[11–14]. The combination of a selfish drive system and pathogen blocking by Wolbachia has led to successful pilot trials for suppression of mosquito population size and replacement of infected mosquito populations so they can no longer transmit pathogens[15–17].

Wolbachia are transovarially transmitted from the mother to offspring[6,18–20]. In some arthropods, including the naturally infected vector species complex Culex pipiens, Wolbachia 'modify' sperm in testes, leading to embryonic lethality if the infected male mates with either an uninfected female or a female harbouring an incompatible Wolbachia strain. When both male and female are infected with the same Wolbachia, the modification is 'rescued', and compatibility is restored[21]. This reproductive alteration, termed 'cytoplasmic incompatibility' (CI), comprises the most common form of Wolbachia-induced parasitism and, as studied in C. pipiens, can lead to highly diverse unidirectional and bidirectional incompatibility phenotypes[22]. CI is currently used in vector control studies for population replacement by Wolbachia-infected strains that block arbovirus transmission[17] or population suppression that reduces the number of mosquito vectors[23]. Notably, the genes responsible for CI[24–26] occur in the eukaryotic association module of Wolbachia's temperate phage WO[25–27], which laterally transfers between Wolbachia coinfections and evolves rapidly[28–32]. Overall, these findings emphasize the importance of further investigating mobile genetic elements in Wolbachia.

While recent studies shed light on the role of phage WO in Wolbachia genome evolution and CI, other extrachromosomal elements, such as plasmids, have not been detected in the symbiont. Notably, over half of the species in the closely related Rickettsia genus have plasmids[33] that play roles in DNA replication, partitioning, mobilization, and conjugation[34,35] and offer a potential tool for genetic manipulation of diverse members of Rickettsia[36,37]. Similar genetic manipulation strategies for Wolbachia are conceivable[27]; however, previous efforts to search for such extrachromosomal mobile genetic elements have not been successful[38,39]. The lack of isolates limit direct insights into Wolbachia genomics, and most metagenomic approaches thus far rely on pooled individuals grown in the laboratory environment due to low infection densities[40,41]. Since these limitations can conceal naturally occurring genomic diversity among Wolbachia populations, highly resolved analyses of individual mosquitoes may reveal additional insights into the Wolbachia 'mobilome', the pool of all mobile genetic elements associated with Wolbachia populations.

Here we sequence ovary samples from four wild-caught C. pipiens individuals captured in Southern France from a single trap. Using genome-resolved metagenomic and pangenomic analysis strategies, we were able to reconstruct and compare near-complete Wolbachia genomes from each individual. Besides a diverse set of virus-associated genes that were missing or absent in the reference Wolbachia genome wPip Pel, our data reveal the first lines of evidence for an extrachromosomal circular element with genetic and functional hallmarks of a plasmid that we tentatively name pWCP.

## Results

**A Wolbachia metagenome-assembled genome (MAG) is recovered in each sample.** Shotgun sequencing of DNA recovered from ovary samples of four C. pipiens individuals (O03, O07, O11, O12) resulted in 65–78 million paired-end sequences after quality filtering. Metagenomic assembly of each sample individually yielded 147K–183K contiguous DNA segments (contigs) >1 kbp, which recruited 48.1–72.9% of the raw sequencing reads. The relatively high fraction of unmapped reads were likely due to challenges associated with the assembly of environmental metagenomes[42,43], especially in the presence of eukaryotic host genomic DNA. Supplementary Table 1 reports statistics for the raw number of reads and assembly results for each sample. We employed a metagenomic binning strategy that uses sequence composition signatures and differential coverage statistics of contigs across samples. For each ovary sample, we were able to reconstruct a highly complete single bacterial MAG that resolved to Wolbachia (Table 1).

**Wolbachia MAGs include highly covered non-phage contigs.** The relatively low number of single-nucleotide variants (0.01–0.05%, Supplementary Table 2) suggested that each Wolbachia MAG represented a nearly monoclonal population of bacterial cells in the ovary metagenomes. In addition, the high average nucleotide identity across our MAGs and the 1482 kbp reference genome for Wolbachia, wPip Pel[40] (99.1–99.98%, Supplementary Table 3) suggested a high degree of conservation between the endosymbionts of different individuals. Our metagenomic read recruitment analyses using Wolbachia MAGs revealed consistent coverage statistics averaging 168×–491× for contigs that were enriched with bacterial genes. However, a subset of contigs in each individual displayed approximately a five-fold increase in coverage (Supplementary Table 4). Because Wolbachia harbour prophage WO that can enter the lytic cycle

**Table 1 Wolbachia MAG estimates**

| MAG | Percent completion (PC) | Percent redundancy (PR) | Number of contigs (N) | Number of genes (n) | Length (total number of nucleotides) | GC content (%) |
|-----|------|------|-----|------|-----------|-------|
| O03 | 94.24 | 0.72 | 93 | 1091 | 1,213,072 | 33.83 |
| O07 | 94.24 | 0.72 | 127 | 1181 | 1,317,313 | 33.78 |
| O11 | 94.24 | 0 | 99 | 1085 | 1,208,099 | 33.84 |
| O12 | 94.24 | 0 | 99 | 1113 | 1,237,800 | 33.95 |

Estimates include completion and redundancy estimates, number of contigs (N), number of genes (n), total number of nucleotides and percentage of GC. More than 90% completion and <10% redundancy based on the single occurrence of 139 single-copy genes (SCG) identified from the collection by authors in ref. [77] suggest high completion of the bins

and form phage particles[27,44,45], we postulated that some of the contigs could be phage associated, which would explain coverage inconsistencies. We used the five prophage regions identified in the *w*Pip Pel reference genome[27,40] to identify contigs enriched with genes of phage origin. Contigs classified and validated through homology searches against phage WO matched to contigs that were highly covered in our MAGs, confirming that most shifts in coverage could be attributed to phages of *Wolbachia*. However, surprisingly, five contigs in our MAGs (Contig O12_A, Contig O11_A, Contig O07_A, Contig O07_A', and Contig O03_A, Supplementary Data 1) showed no homology to prophage WO despite their remarkable coverage that ranged between 720× and 2176×. Interestingly, the 5' and 3' ends of these contigs showed homology to the non-coding flanking regions of *w*Pip's ISWpi12 transposable element (TE; WP0440, WP1209, and WP1347) of the IS110-family[46]. Given their (1) high coverage in metagenomes, (2) lack of homology to prophage WO, and (3) putative association with IS110 TE, we hypothesized that these contigs could represent extrachromosomal elements.

**pWCP: a *Wolbachia*-associated putative plasmid**. Based on homology between the ends of these five contigs and the flanking regions of IS110 TE, we predicted that the missing region was the latter element. We artificially circularized Contig O11_A (8037 bp) and inserted an IS110 TE (1386 bp) based on the overlapping 5' and 3' ends (Fig. 1). Metagenomic read recruitment onto this artificially circularized contig, which we tentatively name 'pWCP' (for plasmid of *Wolbachia* endosymbiont in *C. pipiens*), showed consistent coverage over its entire length except a clear two-fold coverage increase in a region that matched to the IS110 TE in all four *C. pipiens* individuals in our study (Supplementary Figure 1). These data suggested the IS110 TE are located in the extrachromosomal pWCP while some others could be integrated in the bacterial chromosome. The read recruitment from three *C. pipiens* egg metagenomes generated in a previous study[32] confirmed the near identical presence of pWCP in all three, including the increase in coverage matching to the IS110 TE (Supplementary Figure 1).

To validate the circular and extrachromosomal nature of pWCP independently of short-read recruitment- and assembly-based strategies, we generated long reads from additional *C. pipiens* complex samples using a MinION sequencer. Since MinION sequencing occurs with no polymerase chain reaction (PCR) or downstream assembly steps, we hypothesized that long reads that match to pWCP should never be (1) flanked by genomic regions matching to the *Wolbachia* chromosome and (2) longer than the pWCP itself. Our MinION sequencing analysis resulted in 14,808 high-quality sequences that were >5000 nucleotides. While a significant fraction of these reads were eukaryotic contamination and the lambda phage DNA that we used to pad our low-biomass samples (see Methods), a local BLAST search of artificially circularized pWCP sequence against long reads revealed 19 that aligned to pWCP with an *e* value of <1e−20 (Supplementary Data 2). Thirteen of these 19 reads covered >50% of the length of pWCP and contained no other genomic region. In other words, each of the long reads were equal to or shorter than the length of pWCP, as expected for an extrachromosomal element (Fig. 2a). Moreover, 6 of these 13 reads were exactly equal to the length of pWCP, covering its entirety with non-identical start positions, confirming its circularity (Fig. 2a). The final 6 long reads that covered <50% of the length of pWCP had specific matches to the IS110 TE (Fig. 2b); this result is consistent with the multiple occurrences of the IS110 TE in the Wolbachia genome and explains the

increase in coverage in metagenomic read recruitment results (Supplementary Figure 1).

To further validate the extrachromosomal nature of pWCP, we PCR screened genomic DNA from the wild caught individuals as well as from tetracycline (TC)-treated laboratory lines (TC1–TC3; negative controls). TC treatment eliminates *Wolbachia* from its hosts and is commonly used to generate *C. pipiens* uninfected laboratory lines[47]. First, using LCO1490 and HCO2198 mitochondrial cytochrome c oxidase subunit I (COI) invertebrate primers[48], we detected the presence of a ~708-bp band in the four *C. pipiens* individuals and the three *Wolbachia*-free *Culex quinquefasciatus* controls treated with TC (TC1–TC3), confirming the presence of arthropod DNA in all samples (Fig. 1b). Next, we verified the presence of *Wolbachia* DNA in the first four samples by PCR amplifying a 438-bp fragment of the *Wolbachia* 16S rRNA gene using Wspec-F and Wspec-R primers[49]. No band was observed in the TC samples, confirming the absence of *Wolbachia* (Fig. 1c). Critically, a ~1800-bp fragment amplified with primers 263F and 2127R (Supplementary Table 5) designed from the ends of the artificially circularized contig within DnaB gene (and uniquely matching those sites), confirmed the circular nature of the pWCP and its presence only in *Wolbachia*-infected *C. pipiens* samples (Fig. 1d). We also designed primers to Sanger sequence across the circular gap (see Supplementary Table 5 for primer sets EC_1–EC_7), and results confirmed the pWCP sequence obtained with both Illumina and MinION sequencing. Finally, we amplified IS110 TE with primers EC_4F and EC_4R (Supplementary Table 5) in the four *C. pipiens* samples studied herein while observing no band in the *Wolbachia*-free samples (Fig. 1e) in order to verify the strict association of IS110 TE with *Wolbachia* (Fig. 1f). These results were further confirmed using additional rifampicin- and oxytetracycline-treated *C. pipiens* samples (Supplementary Figure 2). Of note, we verified the presence of pWCP in the originating TC1–TC3 and *C. pipiens* laboratory stocks prior to antibiotic treatment.

The average nucleotide identity of the four independently assembled pWCP sequences from individual mosquitoes was 99.65–100% (Supplementary Table 6). In addition, we identified a 8315-bp contig in the *w*Pip JHB assembly (ABZA01000008.1[50]), which also was >99.53% identical to each of the four pWCP sequences (Supplementary Table 6). The IS110 TE was 100% identical across samples and confirmed as clonal. Our additional read recruitment analyses from publicly available metagenomes (SRR5810516, SRR5810517, SRR5810518)[32] also revealed the widespread occurrence of pWCP in *C. pipiens* individuals from Turkey, Algeria, and Tunisia (Supplementary Note 1, Supplementary Figure 3) at a similar 4.2–7.3-fold higher copy number relative to the *Wolbachia* genome. We also confirmed that pWCP and the *Wolbachia* genome display a similar tetranucleotide composition (Supplementary Note 2, Supplementary Figure 4). Overall, these findings suggest that pWCP is maintained over the long term with *Wolbachia* from these strains.

**pWCP encodes an IS110 TE, 14 genes, and an intergenic repeat region**. The IS element is homologous to ISWpi12 of the IS110 family[46] based on IS finder platform search. Annotation of genes in pWCP also revealed the presence of a disrupted *DnaB*-like helicase, two *RelBE* loci, a *ParA*-like gene (each with identical AA sequences across samples, Supplementary Note 3), and multiple genes encoding hypothetical proteins (Fig. 1f, Supplementary Data 3). The *ParA*-like partitioning sequence showed amino acid homology to the chromosome partitioning of plasmid protein (ParA) identified in *Ca. Caedibacter acanthamoebae* (an endosymbiont of acanthamoebae; *e* value: $2 \times 10^{-46}$), the bacterium *Odyssella thessalonicensis* (*e* value: $9 \times 10^{-45}$), and *Rickettsia*

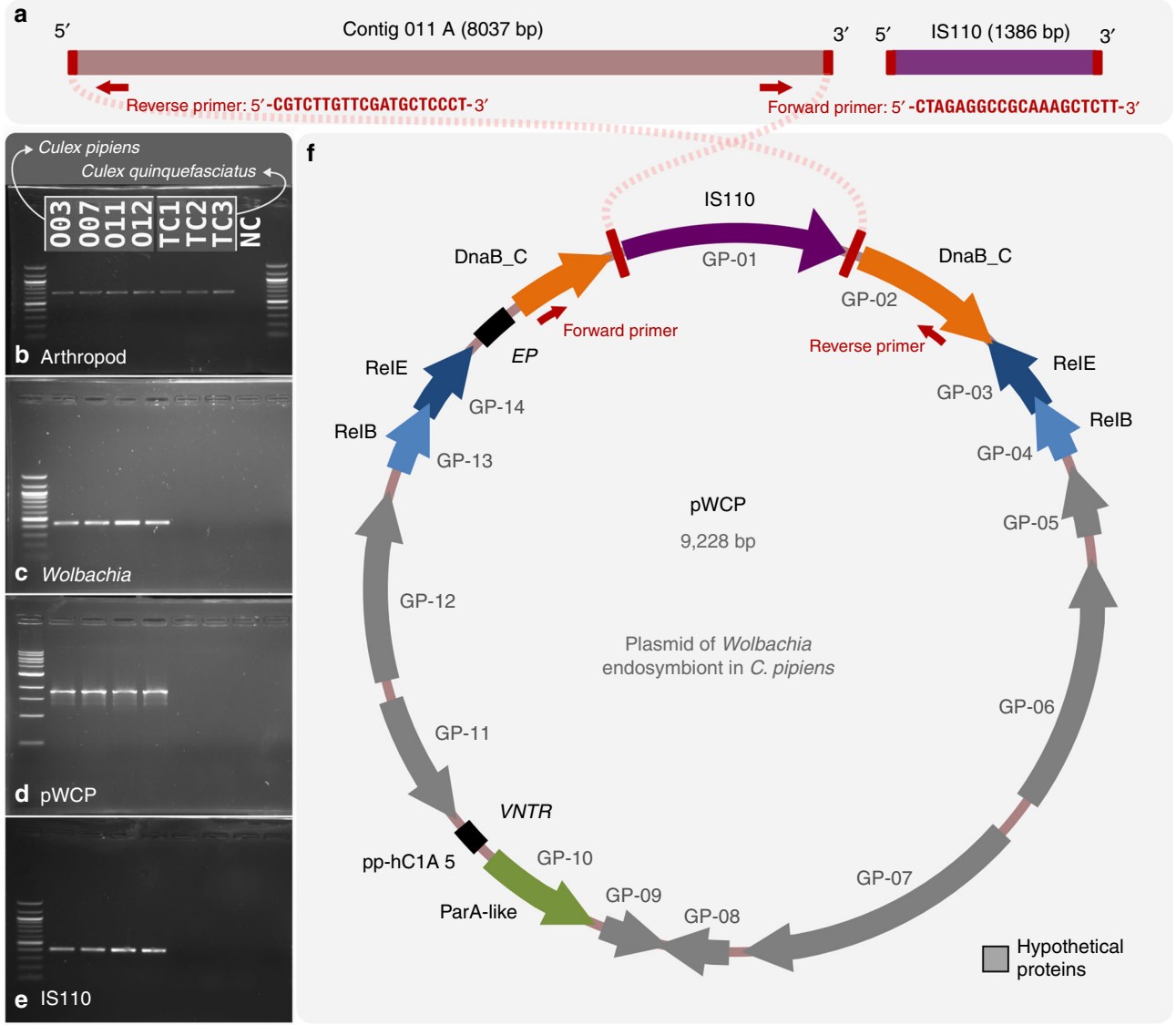

**Fig. 1** The artificially circularized genome of putative plasmid pWCP. **a** illustrates Contig O11_A and IS110 transposable element (TE) identified in our assembly. The red rectangles are regions of 100% nucleotide identity between the two contigs. Outward PCR primers were designed to amplify and confirm circularity of the sequence. **b–e** Gels for PCR tests to confirm a *Wolbachia*-associated circular genome. To verify the presence of arthropod DNA in our four *Culex pipiens* samples and the tetracycline-treated (TC) *Culex quinquefasciatus* samples, we PCR amplified a 708-bp sequence using LCO1490 and HCO2198 primers (**b**). A 438-bp fragment of the *Wolbachia* 16S rRNA gene (**c**), an approximately 1800 bp sequence amplified with the outward primers designed to support circularity of the genome, as illustrated in top panel (**d**) and a 431-bp of IS110 TE (**e**) were obtained in wild *C. pipiens* samples O03-O07-O11-O12 while no amplification was observed in *Wolbachia*-free samples. NC corresponds to negative control. **f** illustrates the complete genome. Each arrow represents an open reading frame (ORF). ORFs with no homology to a known function are shown in grey. *ParA*-like (green), *RelBE* toxin–antitoxin operon (blue), and *DnaB_C* replicative DNA helicase (orange) that is disrupted by the ISWpi12 TE of the IS110 family (purple) are represented by arrows (with an *e* value < e$^{-12}$ from an NCBI Conserved Domain or Pfam Search). Black squares represent the location of (1) the variable number tandem repeat (VNTR) and (2) the extragenic palindrome (EP) region

*raoultii* (*e* value: $7 \times 10^{-41}$). The *RelBE* toxin–antitoxin (TA) locus has been identified in multiple *Wolbachia* genomes[51] and is often associated with prophage WO regions (e.g., of *w*VitA, *w*Ha, *w*Mel, *w*Au, *w*Ri, *w*Suzi, *w*Fol, *w*Inc), specifically within the tail and/or capsid modules. In other bacteria, this TA system can promote the stability of its encoding mobile element, including plasmids or pathogenicity islands, through post-segregational killing of cells that have lost the antitoxin component of the TA operon[52,53]. Most remaining pWCP genes were hypothetical and unique to either *w*Pip and/or the B-*Wolbachia* phyletic supergroup[54] (Supplementary Data 3), including GP-09 and GP-11 which showed a very weak homology to a Transcription Factor

and a Terminase, respectively (*e* value > 0 in SCOPe, Pfam, and Clusters of Orthologous Group (COG), highlighting the need for further functional characterization of this newly discovered mobile genetic element. In particular, terminase proteins bind and package DNA into the capsids of phage particles[55]. These data indicate that the extrachromosomal DNA could potentially fall into three categories: a simple plasmid, a mini-chromosome of *Wolbachia*, or a plasmid-like replicon that hijacks the capsids of phage WO.

Beyond the putative coding regions, alignment of all contigs revealed an intergenic variable number tandem repeat (VNTR) region (Fig. 3a, b, Supplementary Figure 5), characterized as 16-nt

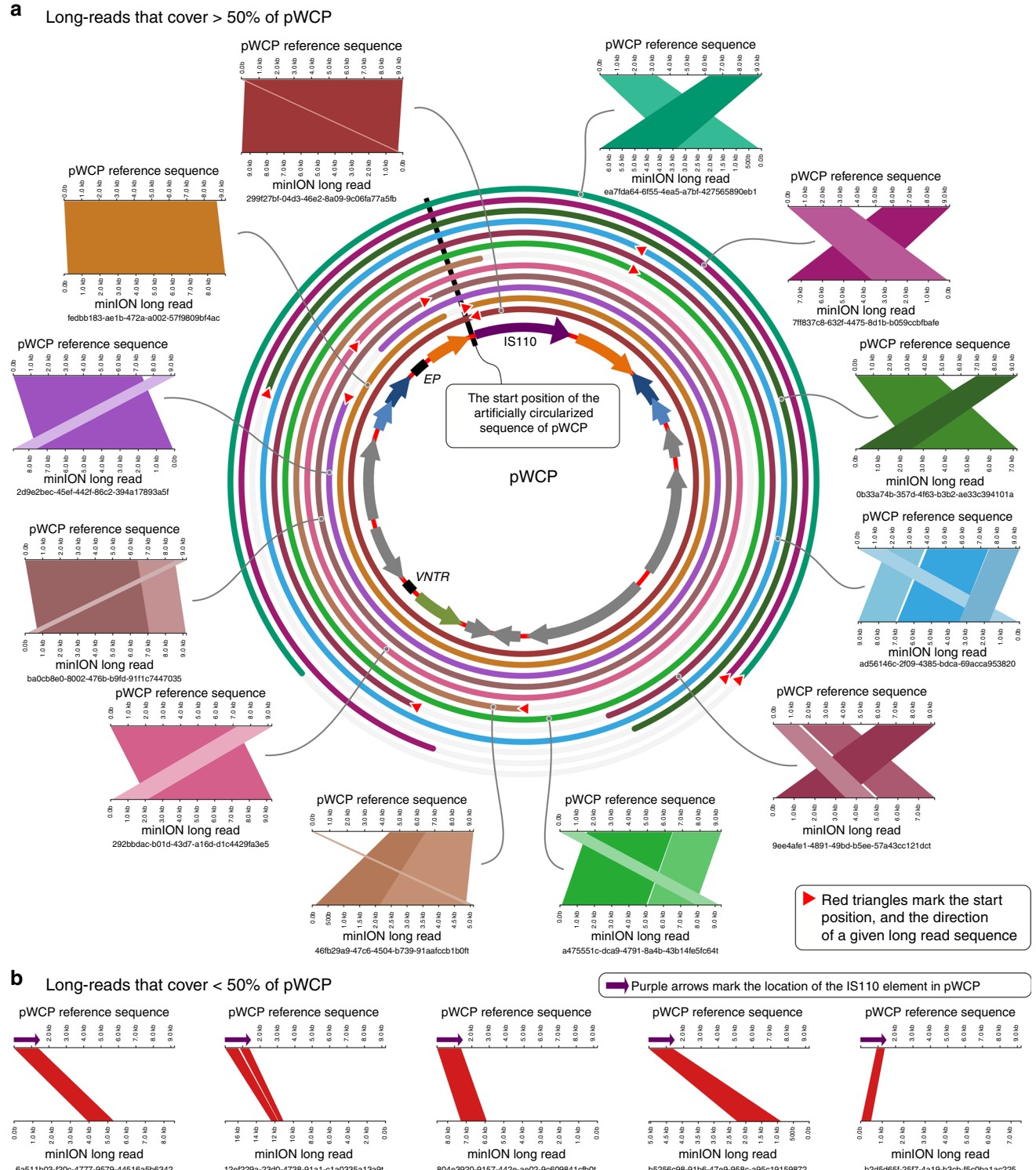

**Fig. 2** Alignment of MinION long reads to pWCP. **a** The alignment of long reads that cover >50% of the pWCP genome (only 12 of 13 total long reads are shown; we omitted 1 from this display solely due to space considerations). Each rectangular figure shows high scoring pairs (HSPs) and their alignments between pWCP and long reads. The broken HSPs that are parallel to each other are due to low-quality regions in long read sequences, and they are shown in different shades. Concentric circles around pWCP demonstrate the alignment of each long read and their start and stop positions. **b** The alignment of long reads that cover <50% of the pWCP genome (only 5 of 6 are shown). Every long read shown in the figure has a hit to the IS110 TE

repeats adjacent to *parA* that vary in number among individual pWCP sequences. Recent studies observed the same repeat region, identified as pp-hC1A_5, and used it to genotype different strains of *Wolbachia*[56,57], yet these have not been studied at an individual level. The authors suggested that a deletion in the

intergenic polymorphic region could serve as a recombination or horizontal gene transfer site[56]. Alternatively, we hypothesize that the direct repeats, as present in iteron plasmids, could indicate a potential origin of replication and play a role in copy number control[58]. Our analysis of the pWCP sequence also revealed a

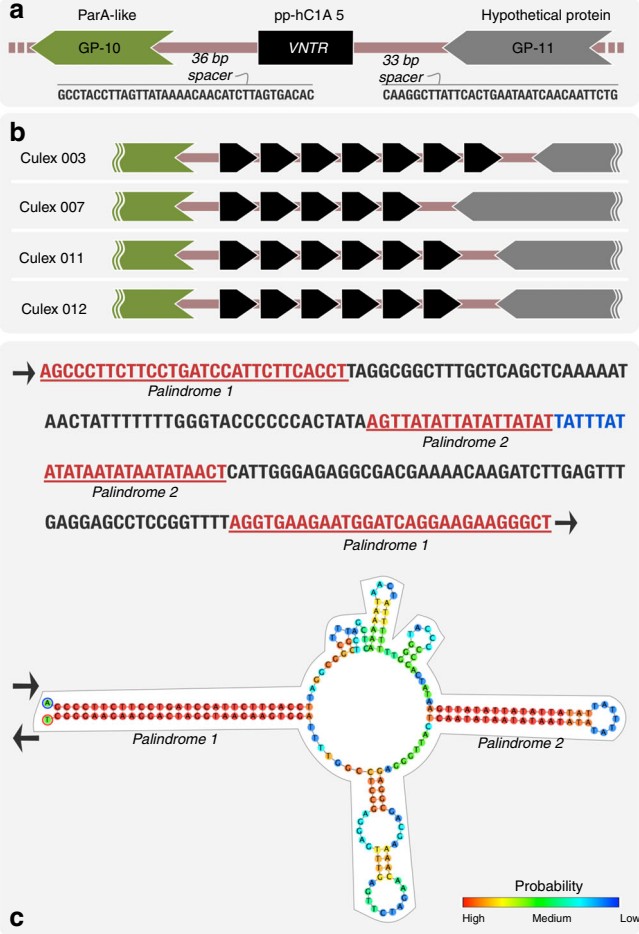

**Fig. 3** pWCP contains a variable number tandem repeat (VNTR) region and extragenic palindrome (EP) sequence. **a** A VNTR region is located between *parA* and uncharacterized gene in the circular genome. While the number of repeats varies across individuals (**b**), the 36- and 33-bp spacers are conserved. Each black arrow represents a 16-nt repeat. The predicted DNA structure of the EP sequence is illustrated in **c** where color indicates probability of each base pairing. Red represents the strongest probability, whereas blue is the lowest

209-bp extragenic palindrome (EP) region with two palindromes (Fig. 3c). Although the role of these sequences is not clear, the closely related plasmid of *Rickettsia monacensis* (pRM) harbours four perfect and four imperfect palindromes[34].

**The *Wolbachia* metapangenome reveals novel viral genes**. The assembly and binning of individual mosquitoes from the wild also enabled comparison of the diversity and the gene content of prophage WO regions in our *Wolbachia* genomes vs. the *w*Pip Pel reference genome. We performed a metapangenomic analysis of the four *Wolbachia* MAGs and *w*Pip Pel in conjunction with the four metagenomes from individual mosquitoes. By linking genes to their abundances in *C. pipiens* metagenomes, we aimed at tying genomic and environmental data. To determine gene coverages, we used the *Wolbachia* MAG O07 as reference for read recruitment since (1) it was the largest MAG in size with most number of genes (Table 1) and (2) all MAGs were over 99.8% identitical (Supplementary Table 3).

The *Wolbachia* pangenome contained 1166 gene clusters (that is, groups of homologous predicted open reading frames (ORFs) based on amino acid sequence identity), the majority of which were conserved across all five genomes (Fig. 4, Supplementary

Figure 6). *w*Pip Pel and *Wolbachia* MAG O07 carried the largest number of unique gene clusters (Supplementary Data 4). Genes that were unique to *w*Pip Pel ($n = 41$) encoded functions including several transposases, bacteriophage capsid protein coding genes, and other phage-related sequences, most of which were associated with known *Wolbachia* prophages.

Gene clusters unique to MAG O07 ($n = 56$) included a gene coding for an ankyrin and tetratricopeptide repeat protein previously identified in phage WO from *Nasonia vitripennis* wasps[27]. Ankyrin and tetratricopeptide repeats mediate a broad range of protein–protein interactions (apoptosis, cell signaling, inflammatory response, etc.) within eukaryotic cells and are commonly associated with effector proteins of certain intracellular pathogens[59,60]. There was also a Retron-type reverse transcriptase and genes coding for Transposases (COG3293 and a Transposase InsO and inactivated derivatives gene). Although most remaining unique O07 gene clusters had no functional annotation, about a third matched to eukaryotic viral genes based on homology searches in the NCBI's non-redundant protein sequence database (Supplementary Data 5).

These data add to previous studies showing that regions of genomic diversity between closely related *Wolbachia* genomes are often virus associated[50,61–63]. Note that most gene clusters with genes that were unique to MAG O07 did recruit reads from the three other metagenomes (Fig. 4, Supplementary Figure 6), suggesting that, even though they were not characterized in our other MAGs, they did occur in *C. pipiens* metagenomes. Absence of these genes from our MAGs are most likely due to (1) assembly artifacts that result in fragmented contigs that are too short to be considered for binning or (2) mutations in the gene context that affect the gene caller to identify them properly. Gene clusters that matched to pWCP also occurred only in our *Wolbachia* MAGs and were missing in *w*Pip Pel (Supplementary Figure 6), which is expected since the *w*Pip Pel reference genome is solely composed of the *Wolbachia* bacterial chromosome[40]. Overall, the metapangenome sheds light on a substantial amount of viral genetic diversity, revealing almost as many virus-associated genes as the ones that were previously recognized in the reference genome *w*Pip.

Unlike our *Wolbachia* MAGs, *w*Pip Pel is a high-quality genome assembled into a single scaffold. Even though *w*Pip Pel is not completely closed[40], it offers a more complete representation of the synteny of genes in comparison to our MAGs. Hence, in addition to ordering gene clusters based on their distribution patterns across all genomes (Supplementary Figure 6), we took advantage of the *w*Pip Pel genome to determine the order of gene clusters in the metapangenome based on the order of *w*Pip genes in the *w*Pip Pel genome (Fig. 4). This 'forced synteny' allowed us to investigate the diversity and abundance of phage genes in the context of the five previously identified prophage WO regions in *w*Pip Pel (Fig. 4). Some gene clusters within prophage regions appeared to be unique to *w*Pip Pel and were not detected in our metagenomes. It is possible that these genes were not recovered from our MAGs due to small contig size (that is, contigs were too short to be considered for binning). However, our MAGs often carried upstream and downstream phage genes in these regions, suggesting that while some phage genes were conserved across all genomes, others differed significantly from their *w*Pip Pel counterparts (Fig. 4, Supplementary Data 4). It is possible that a set of new phage-associated genes only found in MAGs (Fig. 4, Supplementary Data 5) have functional homologues in *w*Pip Pel. Previous studies indeed show WO genes that have distinct nucleotide sequences yet similar predicted functions[30]. However, it is also possible that some genes detected only in our MAGs at the sequence level may also be encoding unique functions compared to known phage genes; for instance, eukaryotic-like homologues were recently shown as constituents of phage WO[27].

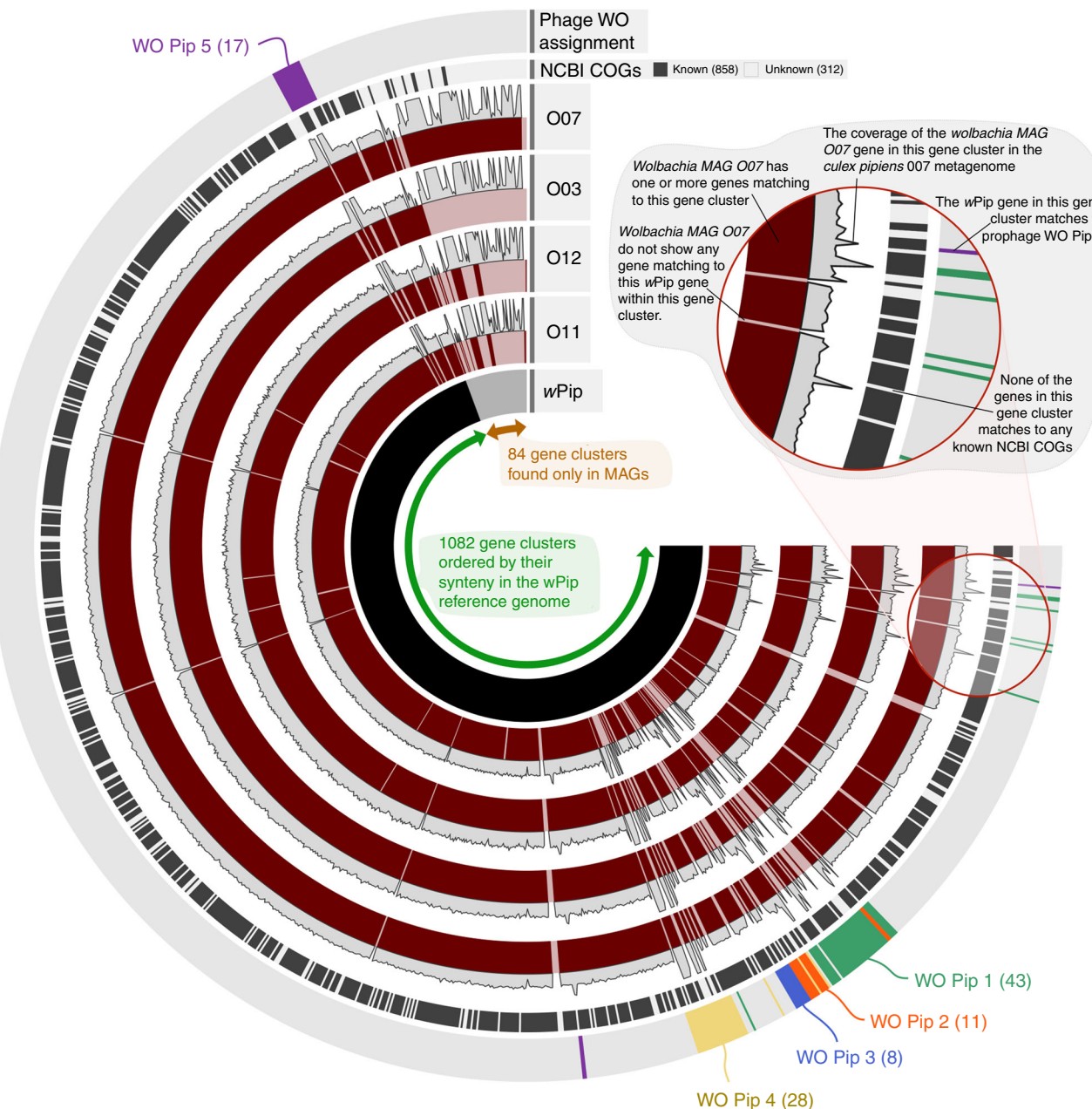

**Fig. 4** *Wolbachia* metapangenome in the context of *w*Pip Pel genome synteny. The figure shows the presence–absence of 1166 gene clusters in the pangenome of four *Wolbachia* metagenome-assembled genomes (MAGs) and the reference genome *w*Pip Pel. The gene clusters (i.e., groups of homologous genes based on amino acid sequence identity) are organized based on *w*Pip Pel synteny. Each MAG is represented by two layers, where the first layer indicates the presence or absence of a gene cluster in a given MAG, and the other shows the average coverage of each Wolbachia MAG O07 gene cluster in the corresponding *C. pipiens* metagenome. The second to last layer shows whether genes in a given gene cluster have a match in NCBI's COGs. The outermost layer associates gene clusters with previously identified prophage regions in the *w*Pip Pel genome. The number of gene clusters assigned to WOPip prophage regions are indicated in parenthesis

Metagenomic read recruitment revealed between a 1.5- and 5-fold increase in coverage between pWCP and some structural phage genes (e.g., WP0415 and WP0446 Tail genes) in our metagenomes compared to the coverage of the bacterial chromosome in all four *C. pipiens* individuals (Supplementary Data 6). The forced synteny organization of gene clusters also revealed that a single prophage WO region could include both high- and low-coverage phage genes (Fig. 4). The multi-copy occurrence of pWCP could explain its increased coverage (Supplementary Figure 1), and differential coverage regimes for genes within a single prophage region could be explained by at

least two different models. First, increased coverage of some prophage genes could be attributed to lytic activity: the prophage genes displaying lower bacterial-like coverage are not part of the virion, while those with higher coverage correspond to phage genes that are replicated and packaged into phage capsids. This lytic model is consistent with the observation of phage particles in *C. pipiens* mosquitoes[44,45], observed lytic activity of *Wolbachia* phages in *Nasonia* testes, and the sequencing of WO genomes from purified phage particles[27]. The partial replication and packaging of prophage WO genes could result from either a 'less than headful' mechanism of packaging, as described in model

phages P1 and T4[64,65], or it could represent active vs. degenerate prophage variants in the wPip Pel chromosome. Second, some genes in prophage WO genomes could be copied throughout the Wolbachia chromosome explaining the increase in coverage due to their multi-copy occurrence. This model is supported by the prophage duplication events observed in the wRi and wSuz genomes[61,66] as well as the presence of TEs within and/or flanking prophage variants[27,40] that could enable genomic rearrangement and duplication. These models may not be mutually exclusive, and the system could involve both duplication of prophage genomes and the induction of phage particles.

## Discussion

Shotgun metagenomes from individual C. pipiens ovary samples allowed us to de novo reconstruct Wolbachia genomes from single mosquitoes and compare these MAGs to each other as well as to the reference wPip Pel genome through pangenomic strategies. Our data reveal an extensive diversity of previously undetected Wolbachia phage WO and other viral genes and notably the first indication that Wolbachia harbours a candidate plasmid, shedding new light on the richness of the Wolbachia mobilome. The definition of a plasmid varies; here we adhere to the typical characterization of a plasmid as a small hereditary, extrachromosomal, circular element.

Even though a Wolbachia plasmid has not been reported before, we did find evidence of its occurrence in C. pipiens metagenomes in previous studies. It is likely that the previous efforts overlooked this element due to computational challenges associated with the assembly of metagenomic short reads. Beyond the IS110 TE that occurred both in the plasmid and Wolbachia chromosome, pWCP contained a region of intergenic VNTR that differed across individuals in our study. This suggests that the co-assembly of pooled individuals may yield fragmented contigs. The same VNTR sequences were found in C. pipiens from Australia, Argentina, USA, Italy, Japan, Israel, and Greece[56], and we confirmed the presence of pWCP in C. pipiens samples from countries of the Mediterranean basin[32]. The high similarity of pWCP sequences across C. pipiens metagenomes in addition to its global prevalence suggest evolutionary constraints and a possible functional role in Wolbachia symbiosis. However, we did not detect the plasmid in other Wolbachia strains through screening available metagenomes from the fly Drosophila melanogaster, the planthopper Laodelphax striatella, and Anopheles gambiae mosquito.

Our work demonstrates that the combination of genome-resolved metagenomics, long-read sequencing, and pangenomic strategies provide an effective computational framework to investigate the diversity and distribution of mobile genetic elements in endosymbionts that are challenging to cultivate. Furthermore, it shows the importance of studying distinct individuals from wild mosquito populations, in parallel with controlled experiments in laboratory settings, to improve our understanding of the Wolbachia mobilome. The fragmented nature of Wolbachia MAGs in our study emphasizes the critical need for harnessing the power of emerging long-read sequencing technologies to characterize complex genomic variations of Wolbachia and its mobilome at finer scales. Wolbachia has been so far recalcitrant to`genetic modification, but the discovery of pWCP and phage WO may create new avenues for effective genome-editing strategies.

## Methods

**Sample collection**. We collected mosquito specimens using a carbon dioxide mosquito trap located in Languedoc, Herault, France in May 2017 (Camping l'Europe de Vic La Gardiole, EID Méditerranée). Specimens were transported alive to the laboratory immediately upon recovery. We anesthetized adult females for 4 min at −20 °C and proceeded to species-level identification. To remove potential contaminants from the insect surface, we gently vortexed specimens for 1 min in 1 ml cold (4 °C) 96% ethanol. We then transferred them to a new 1.5 ml tube with 1 ml sterile cold (4 °C) phosphate-buffered saline (PBS) 1× solution and gently vortexed them again for 10 s to avoid DNA precipitation with ethanol. Finally, we transferred specimens onto a sterile microscope slide with sterile PBS 1× on top of a cold plate and dissected two ovaries from four specimens using sterilized tweezers. We preserved ovary samples at −80 °C until further processing.

**C. pipiens complex controls**. We obtained Wolbachia uninfected mosquitoes by treating the host with antibiotics. For antibiotics treatment we either used TC, which inhibits protein synthesis, or rifampicin, which interferes with nucleic acid synthesis[67]. The TC-treated lines[47] shown in Fig. 1 (C. quinquefasciatus SLAB-TC lines, ISEM, France) were raised at least 1 year without TC in standard laboratory conditions before beginning experiments. The rifampicin and oxytetracycline-treated C. pipiens lines were kindly provided by Maria del Mar Fernandez de Marco (Animals Plant Health Agency, UK).

**Metagenomic library preparation and sequencing**. We extracted total genomic DNA from each ovary sample, hereafter referred to as O03, O07, O11, and O12, using the MoBio PowerFecal DNA Isolation Kit (QIAGEN Inc., Germantown, MD, USA). We used an E220 Covaris instrument (Covaris, Woburn, MA, USA) to sonicate 3.8–5.7 ng of genomic DNA. We end-repaired and 3′-adenylated resulting fragments and used the NEBNext Ultra II DNA Library Prep Kit for Illumina (New England Biolabs, Ipswich, MA, USA) to add NEXTflex PCR-free barcode adapters (Bioo Scientific, Austin, TX, USA). We purified ligation products by Ampure XP (Beckman Coulter, Brea, CA, USA) and PCR-amplified DNA fragments (>200 bp; 2 PCR reactions, 14 cycles) using Illumina adapter-specific primers and NEBNext Ultra II Q5 Master Mix (NEB). After library profile analysis using an Agilent 2100 Bioanalyzer (Agilent Technologies, Santa Clara, CA, USA) and quantitative PCR quantification using the KAPA Library Quantification Kit for Illumina Libraries (KapaBiosystems, Wilmington, MA, USA), we sequenced the library using a HiSeq4000 Illumina sequencer (Illumina, San Diego, CA, USA) at the Genoscope in Evry, France, generating 151 bp paired-end reads. To remove the least reliable data, we filtered the raw sequencing results using cluster intensity and chastity filter as described in ref. [68].

**Metagenomic assembly and binning**. We used illumina-utils v1.4.4[69] to quality-filter raw paired-end reads with the program 'iu-filter-quality-minoche' with default parameters, IDBA_UD v1.1.2[70] to assemble paired-end reads into contigs, and Bowtie2 v2.2.9[71] for all read recruitment analyses. We processed our contigs that are >1000 nts and read recruitment results with anvi'o v5[72] to recover MAGs from C. pipiens metagenomes. Briefly, we used the program 'anvi-gen-contigs-database' to generate anvi'o contigs databases for each four individual assemblies, during which anvi'o calculates and stores tetranucleotide frequency values and Prodigal v2.6.3[73] identifies ORFs in each contig. We ran default anvi'o HMM profiles on resulting contigs databases using the program 'anvi-run-hmms' and assigned functions to genes using the program 'anvi-run-ncbi-cogs', which searches gene amino acid sequences against the December 2014 release of the COG database[74] using blastp v2.3.0+[75]. We profiled the short reads of our contigs recruited from each four individual metagenomes onto the four assemblies using the program 'anvi-profile', which generates anvi'o profile databases that store the coverage and detection statistics of each contig within each sample independently. We then merged resulting anvi'o profile databases for each sample using the program 'anvi-merge'. For an initial coarse binning, we used the CONCOCT[76] algorithm through the program 'anvi-cluster-with-concoct' and confined the number of clusters to five using the parameter '−num-clusters-requested 5'. We then used the program 'anvi-script-get-collection-info' to identify bins with a bacterial population genome, manually refined them to identify bacterial genomes using the program 'anvi-refine', and assigned taxonomy to resulting MAGs using NCBI's non-redundant protein sequence database with amino acid sequences of core genes. The program 'anvi-refine' allows the identification and refinement of population genome bins through an interactive interface by offering a number of tools including (1) guiding hierarchical clustering dendrograms of contigs (including one based on tetranucleotide frequencies and differential coverage statistics across samples), (2) real-time completion and redundancy estimates of binned contigs based on bacterial single-copy core genes[77], (3) interfaces that display nucleotide-level coverage statistics per gene and contig, (4) functional annotations and synteny of genes, and (5) online sequence homology search options for gene and contig sequences. These tools collectively help minimize the inclusion of non-target contigs (such as eukaryotic contamination) in final bins, especially in complex metagenomes[78]. Except for the manual refinement step, we automated the processing of our raw sequencing data with a snakemake[79] workflow anvi'o v5 implements (http://merenlab.org/2018/07/09/anvio-snakemake-workflows/), and the Code availblility section reports necessary configuration files to fully reproduce this analysis.

**Sample preparation and long-read sequencing**. To investigate the circularity of pWCP, we prepared for a long-read sequencing strategy. We extracted and pooled

DNA from the ovaries of 7 *C. pipiens* and 15 *C. quinquefasciatus* mosquitoes. As the total DNA quantity from this step was only 5.83 ng, we also extracted DNA from the abdomens of 15 *C. quinquefasciatus* that left us with a total of 76.63 ng. To acquire high-molecular weight DNA from these low-biomass samples, we used an in-house Phenol Chloroform protocol for total genomic DNA extraction. Briefly, we used a chloridric acid-washed pestle adapted to the size of a microcentrifuge tube to gently grind tissue in cetyltrimethylammonium bromide buffer (10 ml) mixed with betamercaptoethanol (20 μL) that we incubated for 15 min at 60 °C. We then added 200 μL of phenol:chloroform:isoamyl alcohol and incubated each microcentrifuge tube at room temperature while rotation shaking for 5 min. To minimize inclusion of proteins, we centrifuged the tubes for 10 min at 8000 rpm (5939 × *g*), removed the aqueous phase (the top layer), and placed it into fresh tubes. We repeated those steps with chloroform:isoamyl alcohol and lastly with isopropanol. We incubated microcentrifuge tubes for 10 min at room temperature and mixed by inversion every 3 min. We then centrifuged those tubes for 10 min at 10,000 rpm (9279 × *g*), removed the top layer, and kept the sole DNA at the bottom of the microcentrifuge tube. We next washed DNA with 400 μL of freshly prepared 70% ethanol and centrifuged tubes for 10 min at 10,000 rpm (9279 × *g*). Finally, we kept microcentrifuge tubes in a speed vacuum for 10 min and eluted DNA with 20 uL of molecular grade water. During all laboratory steps, we performed smooth and slow pipetting with cut pipette tips to avoid further shearing the DNA molecules. To prepare mosquitoes' total DNA for sequencing, we used the Rapid Barcoding Kit (SQK-RBK004) by Oxford Nanopore Technologies. We followed the manufacturer protocols, except for one additional step, in which we supplemented samples that did not meet the manufacturer DNA input mass recommendations (400 ng) with linear double-stranded lambda DNA (New England Biolabs) for 'padding'. We then used 1× Agencourt AMPure XP beads (A63882, Beckman Coulter) for sample clean-up and concentration of pooled barcoded samples. A MinION with a pristine R9.4/FLO-MIN106 flow cell (Oxford Nanopore Technologies) sequenced the final prepared library with a starting voltage of −180 mV and a run time of approximately 20 h. While the flow cell had 1293 active pores prior to the run, only 200 were active at the beginning of the run, dropping down to 2 active pores by 20 h. We were not able to determine whether the low number of initial pores and their rapid decline was due to remaining chemical residues from our in-house Phenol Chloroform extraction protocol or a defective flow cell. We used MinKNOW (v1.15.4) for live base-calling and Albacore (v2.1.7) to de-multiplex our samples and convert raw FAST5 files to FASTQ files. For downstream analyses, we only used reads with a minimum quality score of 7.

**Pangenomic analysis and the metapangenome**. Our analysis of the *Wolbachia* pangenome followed the workflow outlined in ref. [80] and at http://merenlab.org/2016/11/08/pangenomics-v2/. Briefly, we first generated an anvi'o genomes storage database using the program 'anvi-gen-genomes-storage' for our four *Wolbachia* MAGs and the *w*Pip reference genome *w*Pip Pel[40] (NCBI Accession ID NC_010981.1). We then used the program 'anvi-pan-genome' using the flag '–use-ncbi-blast', and parameters '–minbit 0.5', and '–mcl-inflation 5'. This program (1) uses the blastp program all vs. all to create a graph of similarities between pairs of amino acid sequences, (2) removes bad hits using the 'minbit heuristic', (3) uses the Markov Cluster algorithm to determine gene clusters, (4) calculates the occurrence of gene clusters across genomes and the total number of genes they contain, (5) realizes a hierarchical clustering analysis for gene clusters (based on their distribution across genomes) and for genomes (based on shared gene clusters), and finally (6) generates an anvi'o pangenomic database that stores all results for downstream analyses. We used the program 'anvi-display-pan' to visualize gene clusters and their distribution across the five genomes in anvi'o interactive interface and summarized our manual selection of gene clusters using the program 'anvi-summarize'. To identify 'phage-like' gene clusters, we used blastn to search for genes homologous to the five phage regions[27] in the *w*Pip Pel genome (hereafter referred to as WOPip1–5). We finally used a custom R script (see the Code availability section) to recover coverage values of each gene cluster using our four metagenomes. We visualized the metapangenome using the program 'anvi-display-pan'.

**Computing the density of single-nucleotide variants per genome and metagenome**. To infer the extent of heterogeneity in metagenomes for each *Wolbachia* MAG, we used the anvi'o program 'anvi-gen-variability-profile' to recover single-nucleotide variants from each merged profile without any filters (following the tutorial at http://merenlab.org/2015/07/20/analyzing-variability/). We then kept only single-nucleotide variants that fall into the context of complete gene calls to minimize erroneous variants that often emerge from mappig artefacts around beginnings and ends of contigs and calculated the percentage of nucleotide positions in each genome that was not clonal in the metagenome.

**Computing the average nucleotide identity between genomes**. To compute the level of similarity between the four *Wolbachia* MAGs and the *w*Pip Pel reference genome, we used the program 'anvi-compute-ani', which called PyANI v0.2.7[81] in 'ANIb' mode to align 1 kbp long fragments of the input genome sequences with the NCBI's blastn program to summarize the average nucleotide identity and alignment coverage scores.

**PCR amplification of DNA**. We used a set of specific primers on the four *C. pipiens* individuals and three *Wolbachia*-free *C. quinquefasciatus* controls generated by TC treatment using standard techniques. The presence of arthropod DNA was verified by amplifying a ca. 708-bp fragment of the Cytochrome Oxidase I gene from arthropod mitochondria using LCO1490: (5'- GGT CAA CAA ATC ATA AAG ATA TTG G -3') and HCO2198: (5'- TAA ACT TCA GGG TGA CCA AAA AAT CA -3') primers[48]. We checked for the presence of *Wolbachia* DNA by PCR amplifying a ca. 438-bp fragment of the *Wolbachia* 16S rDNA gene using Wspec-F: CAT ACC TAT TCG AAG GGA TAG and Wspec-R: AGC TTC GAG TGA AAC CAA TTC primers[49]. The PCR conditions for both sets of primers included a temperature regime of 2 min at 94 °C, followed by 29 cycles of 30 s at 94 °C, 45 s at 49 °C, 1 min at 72 °C, and a final elongation of 10 min at 72 °C. To detect the circularity of the plasmid-like genome, we designed new primers (listed in Results) and used the following cycling parameters: 2 min at 95 °C, followed by 34 cycles of (94 °C for 30 s, 58 °C for 45 s, and 72 °C for 2 min) with a final elongation of 5 min at 72 °C. Finally, to confirm the presence of the IS110 TE, we used primers described in the Results section under the following PCR conditions: 2 min at 95 °C, followed by 34 cycles of (94 °C for 30 s, 55 °C for 45 s, and 72 °C for 30 s) and a final elongation of 5 min at 72 °C. Amplifications by PCR were performed using Platinium Taq DNA Polymerase High Fidelity (5 U/μL, Invitrogen).

**Plasmid characterization**. We used the Glimmer software within Geneious v8.1.9 (Biomatters, Ltd) to identify ORFs and used the NCBI databases for non-redundant protein sequences and conserved domains[82], SMART[83], and HHpred[84] (including SCOPe, Pfam, and COG) to manually annotate putative functions based on amino acid sequence homology searches. We used the IS finder platform (https://isfinder.biotoul.fr/) for IS homology search. The nucleotide analysis plugin in Geneious for EMBOSS identified and illustrated the EP in the plasmid.

**Figures**. We visualized BLAST hits between pWCP and long read sequences using Kablammo[85] (http://kablammo.wasmuthlab.org/) and finalized all figures using the open-source vector graphics editor Inkscape (https://inkscape.org/).

**Reporting summary**. Further information on experimental design is available in the Nature Research Reporting Summary linked to this article.

**Code availability**. The URL http://merenlab.org/data/2019_Reveillaud_and_Bordenstein_et_al_Wolbachia gives access to a reproducible bioinformatics workflow document and ad hoc scripts used for all computational analyses.

## Data availability

The raw sequencing data for shotgun metagenomes are available in the European Nucleotide Archive via accession code PRJEB26028. Sequencing data for the MinION run is available at https://doi.org/10.6084/m9.figshare.7306784. We also made available FASTA files for individual metagenomic assemblies (https://doi.org/10.6084/m9.figshare.6263867), the four metagenome-assembled *Wolbachia* genomes (https://doi.org/10.6084/m9.figshare.6292040), artificially circularized individual plasmid sequences (https://doi.org/10.6084/m9.figshare.6380015), as well as anvi'o merged profile databases (https://doi.org/10.6084/m9.figshare.6263876) and anvi'o files for the *Wolbachia* pangenome (https://doi.org/10.6084/m9.figshare.6291650).

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

## Acknowledgements

We thank Albane Marie, Gregory Lambert, and Jean-Baptiste Panchau from EID (Entente Interdépartementale de Démoustication) for field *Culex pipiens* samples; Maria del Mar Fernandez de Marco from Animals Plant Health Agency for antibiotic-treated *Culex pipiens* individuals; Evan Kiefl, Richard Fox, and Frederick Gavory for their help with computational matters; Phoebe Rice, Sean Crosson, and Marie-Agnès Petit for their insights into plasmid biology; Nick Loman, Loïs Maignien, Manon Bonneau, and Emmanuelle Jousselin for their insights into sequencing. This research was funded by the National Institute for Agronomical Research (INRA) Young Researcher grant (2017) to J.R. and National Institutes of Health awards R01 AI132581 and R21AI133522, National Science Foundation award 1456778, and Vanderbilt Microbiome Initiative funding to Seth R Bordenstein. The INRA Metaprogram (Microbial Ecology and Metagenomics) MEM from INRA provided a USA mobility grant to J.R., and A.M.E. was funded by the University of Chicago start-up funds and the Keck Foundation. K.L. was supported by a Gastrointestinal Research Foundation grant. The funders had no role in study design, data collection and analysis, decision to publish or preparation of the manuscript.

## Author contributions

J.R. conceived the study, performed laboratory assays, analysed the data, prepared figures and tables, and wrote the manuscript; Sarah R. Bordenstein analysed the data, initially identified pWCP, performed laboratory assays, prepared figures and tables, and wrote the manuscript; C.C., M.W., P.M., and I.R. performed laboratory assays; A.S. and O.C.E. provided data analysis tools; K.L. and A.R.W. performed biological assays and long-read sequencing; Seth R. Bordenstein supervised the research and wrote the manuscript; A.M.E. provided data analysis tools, supervised the research, analysed the data, prepared figures and tables, and wrote the manuscript. All authors edited and approved the final draft.

## Additional information

**Competing interests:** The authors declare no competing interests.

