## [Peer Review File · Nature Communications]

Reviewers' comments:

Reviewer #1 (Remarks to the Author):

The manuscript by Reveillaud et al. entitled "The Wolbachia mobilome in *Culex pipiens* includes a putative plasmid" entails metagenomic sequencing and analysis of four mosquito samples that all contain sequences from the endosymbiotic bacterium Wolbachia. The main finding is the discovery of a previously undescribed putative plasmid in the four samples and in additional published sequence datasets from *Culex* mosquitoes.

Overall the manuscript is well written, the methods are clearly explained and all data as well as analyses pipelines are made available.

Although I believe that the plasmid could be real, I think there are some things that needs to be improved and/or clarified for this specific finding:

1. Are the primers 263F and 2,127R in Figure 1d uniquely matching the sites?

I am not able to tell if this DnaB gene is unique to the plasmid, since there are some inconsistencies in the supplementary text regarding the DnaB copies found in three gene clusters. Basically, I'm not 100% sure which cluster contain the DnaB gene from the putative plasmid and if the cluster contain wPip or only MAGs. This needs to be clarified in the supplementary text, but the uniqueness of DnaB should be clarified also in the main text.

2. I assume that the primers used in Figure 1e are not unique to the plasmid, given the higher coverage of this region and the fact that they are inside the transposon. Hence, it is a bit unclear to me what the authors wants to show with this since this doesn't prove any circularization? Or did they run PCR with these primers on the PCR product from Figure 1d?

3. The authors have used tetracycline treated mosquitoes as control for the PCRs in Figure 1. This is a good idea as it shows that you don't get bands from the mosquito genome itself (although it is not exactly the same mosquito species) or from gut bacteria of the lab reared mosquito. However, I think the authors should show that the same mosquitoes that are antibiotics treated actually give positive results before tetracycline treatment, i.e. when you remove Wolbachia do you remove the putative plasmid?

4. The read mapping shown in S. Figure 2, and referred to on p. 6 does not really contain evidence for the circularization of the contig. It just shows that each end of the contig is likely connected to an IS110 element. However, these could in theory be two separate elements, one at each end, since the element is too long for any read pairs to be anchored in the unique sequence on either side. From coverage the IS110 element looks to be present in at least double copies compared

to the rest of the contig. So one possibility is that the whole contig is a tandem duplication (or triplication or more) flanked by IS-elements and anchored inside the chromosomal genome itself. This possibility would also generate a positive result for the primers in Figure 1d. Are there other contigs with similarity to IS110? If there are, the authors should test with for example PCR if the putative plasmid could be anchored in the genome as a tandem repeat.

5. I don't see the rationale behind the paragraph about the insertion of another IS element on p. 6 and 7. Was there any indication that this might be the case? Are there other contigs with IS110 and this element next to each other? The authors do not show the data anyway, so I think the text should be removed unless there is a clear reason for including it.

The pangenome analysis is good and mainly shows similar results as in previous studies, i.e. that phage associated genes are the genes that vary the most between Wolbachia genomes. What might be worth noting and adding to the text for this part is that the reference genome of wPip was never completely closed. Even though it is represented as one contig, it still has a gap and is just artificially joined between two prophage copies (2 and 3) and the genes WP0322 and WP0323.

Reviewer #2 (Remarks to the Author):

Reveillaud, Bordenstein, and colleagues use genome-resolved metagenomics to untangle four metagenome-assembled genomes (MAGs) from *Culex pipiens* individuals that resolved to Wolbachia. The authors analyzed and interpreted these genomes using an array of genomic and pangenomic approaches (implemented in Anvi'o and other software), and found many phage-associated genes and also first evidence for a plasmid, coined pWCP. Overall, the scientific need for this study is apparent, its design is well motivated, the results are clearly presented and backed up by various data, the methods reproducible at sight, and the conclusions sound. Good job!

Major:

- Having supplementary tables as figures is sub-optimal and, somewhat surprisingly, I couldn't find the underlying data on <http://merenlab.org/data/> either. The URL given in the manuscript is merely a placeholder (albeit with a cute kitty). This must be fixed prior to publication.

Minor:

- From the introduction (and as a non-Wolbachia/mosquito-expert), it is not immediately clear why *Culex pipiens* was selected as a host and why it is suitable. The authors mention *Culex pipiens* only once before, close to some *Drosophila* stuff, which was a bit confusing at first. I suggest to explain or motivate this better, 1-2 sentences should suffice.

- The authors report that "only" 50-70% of the metagenomic reads are contained in the assembly (line 121). This is nothing to worry about but, in fact, the norm. Nevertheless, the authors should explain the "high" percentage of unmapped reads to readers not familiar with the "poor" state-of-the-art in metagenome assembly. Suitable references putting their "low" mapping rates into context are: Sczyrba et al., *Nat Methods* 2017 (COI: I'm a co-author of this benchmarking study); Delmont et al., *Nat Microbiol* 2018; ...

- Occasionally, the language is overly complicated and could be simplified, e.g. line 62: "concomitant to" could be replaced by "due to" (which might spare non-native readers the lookup in a dictionary). I also encourage the authors to critically re-read their manuscript and break up stray long and nested sentences (e.g. line 64: "Over the last decade [...]" could easily be split up into 2-3 sentences). However, I wish to emphasize that the authors did a great job already and this last point is largely nitpicking.

Signed,

Andreas Bremges

Reviewer #3 (Remarks to the Author):

This is a well written and state-of-the-art analysis of a very surprising finding, that of plasmids in the *Wolbachia* cells of individual wild captured *Culex pipiens*. I can find no major criticisms of the manuscript, but do have some minor suggestions as listed below

Minor comments:

Please review for the use of "transposasee" and "transposon"; I think sometimes the latter is what is meant, not the former.

Abstract; line 44: can delete "using state of the art... strategies".

Line 48: commas needed after pWCP and ...C. pippiens

Line 63-64: are a current focus. Otherwise it sounds like THIS study)

Line 108; sequenced

Line 115: add Plasmid of.... (name)

Line 17: add shotgun ILLUMINA sequencing

Line 12: do you need "back" ?

Lines 139-143: I think this is very interesting; to my knowledge this had not been done for individuals from the same population to look at SNPs (variants) in Wolbachias; if so it might need to be referenced.

Lines 147-....: increased individual sequence ("gene") coverages could have been due to LGTs, as the Hotopp lab has shown; however your data down the line is very convincing otherwise.

Line 207: mention what antibiotics do and why you used it (its in the Methods, however)

Line 253: ...pWCP and the Wolbachia genome.....

Lines 314-315: a bit confusing; maybe detail this sentence a bit more

Line 344: accessory genes?

Line 361:from our OTHER MAGS

Reviewer #1 (Remarks to the Author):

The manuscript by Reveillaud et al. entitled “The *Wolbachia* mobilome in *Culex pipiens* includes a putative plasmid” entails metagenomic sequencing and analysis of four mosquito samples that all contain sequences from the endosymbiotic bacterium *Wolbachia*. The main finding is the discovery of a previously undescribed putative plasmid in the four samples and in additional published sequence datasets from *Culex* mosquitoes.

Overall the manuscript is well written, the methods are clearly explained and all data as well as analyses pipelines are made available.

Although I believe that the plasmid could be real, I think there are some things that needs to be improved and/or clarified for this specific finding:

1. Are the primers 263F and 2,127R in Figure 1d uniquely matching the sites?

This is correct. Primers 263F and 2,127R in Figure 1d uniquely match the sites. We investigated their specificity by BLAST-searching them against the *Wolbachia* (wPip Pel) chromosome. The best hit for 263F had an e-value of 0.59 and covered only a 20 nucleotide-long section of the genome with a little over 50% identity. Meanwhile 2,127R did even worse with an e-value of 2.3. In addition, we designed seven sets of sequencing primers (Supplementary Table 7) to Sanger sequence across the region. Sequencing results were specific to, and confirmed the occurrence of, the pWCP.

I am not able to tell if this DnaB gene is unique to the plasmid, since there are some inconsistencies in the supplementary text regarding the DnaB copies found in three gene clusters. Basically, I'm not 100% sure which cluster contain the DnaB gene from the putative plasmid and if the cluster contain wPip or only MAGs. This needs to be clarified in the supplementary text, but the uniqueness of DnaB should be clarified also in the main text.

We thank the reviewer for pointing out the confusion and inconsistencies in the Supplementary Material. We corrected this accordingly in our revision. There were three copies of the DnaB gene. One ('bacterial-like') copy was shared between wPip and the MAGs we reconstructed herein. The other two ('plasmid-like') were unique to our MAGs. These could not be aligned as they were too divergent, possibly reflecting different regions of the proteins. These data are in agreement with the disrupted DnaB gene described in the main text.

2. I assume that the primers used in Figure 1e are not unique to the plasmid, given the higher coverage of this region and the fact that they are inside the transposon. Hence, it is a bit unclear to me what the authors wants to show with this since this doesn't prove

any circularization? Or did they run PCR with these primers on the PCR product from Figure 1d?

It is correct that the internal IS primers EC_4F and EC_4R in Figure 1e are not unique to the plasmid. We used these primers to initially confirm the (i) presence of the IS110 transposase in the four *Culex pipiens* samples in our study, and (ii) absence of IS110 in the *Wolbachia*-free samples. We agree with the reviewer that this PCR experiment alone does not prove circularization (for which we generated more data described below), but we believe this experiment was helpful as a first step to confirm the association of the IS to *Wolbachia*.

3. The authors have used tetracycline treated mosquitoes as control for the PCRs in Figure 1. This is a good idea as it shows that you don't get bands from the mosquito genome itself (although it is not exactly the same mosquito species) or from gut bacteria of the lab reared mosquito. However, I think the authors should show that the same mosquitoes that are antibiotics treated actually give positive results before tetracycline treatment, i.e. when you remove *Wolbachia* do you remove the putative plasmid?

We agree with the reviewer that it is important to justify the presence of the putative plasmid in the control *Culex* individuals before antibiotic treatment. We have now verified that infected individuals of *Culex quinquefasciatus* and *Culex pipiens* (including the ancestral lines of both species used for antibiotic treatment) give positive results, and we have now added a sentence in our revision.

4. The read mapping shown in S. Figure 2, and referred to on p. 6 does not really contain evidence for the circularization of the contig. It just shows that each end of the contig is likely connected to an IS110 element. However, these could in theory be two separate elements, one at each end, since the element is too long for any read pairs to be anchored in the unique sequence on either side. From coverage the IS110 element looks to be present in at least double copies compared to the rest of the contig. So one possibility is that the whole contig is a tandem duplication (or triplication or more) flanked by IS-elements and anchored inside the chromosomal genome itself. This possibility would also generate a positive result for the primers in Figure 1d. Are there other contigs with similarity to IS110? If there are, the authors should test with for example PCR if the putative plasmid could be anchored in the genome as a tandem repeat.

We thank the reviewer very much for bringing up this excellent point. Careful consideration of their concern convinced us that we indeed could not rule out the possibility of the “tandem repeats on the chromosome” hypothesis using short reads with assembly-based strategies alone. To address the reviewer's concern, we extracted and sequenced DNA from 37 mosquito samples of the *Culex pipiens complex* using long-read minION sequencing data. The Oxford Nanopore technology reports the sequencing of single molecules with no PCR or assembly step. We hypothesized that if the pWCP is a *bona fide* extrachromosomal and circular element, none of the single molecules matching to it would be flanked by genomic regions that match to the *Wolbachia* bacterial chromosome, and none of the single molecules would be longer than the

pWCP itself. The minION data analysis resulted in 14,808 sequences that passed the intrinsic quality control of the sequencer and that were longer than 5,000 nucleotides. While a significant fraction of these reads were eukaryotic contamination and the lambda phage DNA which we used to 'pad' our low-biomass samples for sequencing (described in detail in updated Methods), BLAST search of the predicted pWCP sequence against these long reads revealed 19 single molecule sequences that aligned to pWCP with an e-value of $<1e-20$. 13 of these 19 reads covered $>50\%$ of pWCP, and contained no other genomic region as each one of them were equal to or shorter than the length of pWCP, confirming its extrachromosomal nature. Notably, the remaining 6 of these 19 long reads which were $<50\%$ of the pWCP length had matches only to the IS110 *Wolbachia* element, as predicted.

We now have updated our Results and Methods sections as well as the reproducible workflow, made raw MinION reads publicly available, included an additional supplementary table, and summarized the alignments of long-reads to the artificially circularized pWCP sequence with the new figure down below.

We thank the reviewer again for bringing up this excellent point, which resolved a critical question in the study. Meanwhile we learned a lot about long-read sequencing, tested a protocol for DNA extraction and sequencing of low biomass samples, and described our experience in detail for similar future attempts.

(a) Long-reads that cover more than 50% of pWCP

(b) Long-reads that cover less than 50% of pWCP

The new Figure 2 shows the alignment of long reads to pWCP. (a) The alignment of long reads that cover more than 50% of the pWCP genome (only 12 of 13 total long reads are shown; we omitted one from this display solely due to space considerations, however all data are reported in the new Supplementary Table). Each rectangular figure shows high scoring pairs (HSPs), and their alignments between pWCP and long reads. The broken HSPs that are parallel to each other are due to low-quality regions in long read sequences, and are shown in different shades. Concentric circles around pWCP demonstrate the alignment of each long read and their start and stop positions. (b) The alignment of long reads that cover less than 50% of the pWCP genome (only 5 of 6 are shown). Every long read shown in the figure has a hit to the IS110 element, and the remainder of these long reads match to the *Wolbachia* chromosome.

5. I don't see the rationale behind the paragraph about the insertion of another IS element on p. 6 and 7. Was there any indication that this might be the case? Are there other contigs with IS110 and this element next to each other? The authors do not show the data anyway, so I think the text should be removed unless there is a clear reason for including it.

We agree with the reviewer. We have now removed the confusing paragraph.

The pangenome analysis is good and mainly shows similar results as in previous studies, i.e. that phage associated genes are the genes that vary the most between Wolbachia genomes. What might be worth noting and adding to the text for this part is that the reference genome of wPip was never completely closed. Even though it is represented as one contig, it still has a gap and is just artificially joined between two prophage copies (2 and 3) and the genes WP0322 and WP0323.

We thank the reviewer for pointing this out. We modified the text to mention this.

We are very thankful for the reviewer's comprehensive evaluation and critical suggestions that strengthened our study.

Reviewer #2 (Remarks to the Author):

Reveillaud, Bordenstein, and colleagues use genome-resolved metagenomics to untangle four metagenome-assembled genomes (MAGs) from *Culex pipiens* individuals that resolved to *Wolbachia*. The authors analyzed and interpreted these genomes using an array of genomic and pangenomic approaches (implemented in Anvi'o and other software), and found many phage-associated genes and also first evidence for a plasmid, coined pWCP. Overall, the scientific need for this study is apparent, its design is well motivated, the results are clearly presented and backed up by various data, the methods reproducible at sight, and the conclusions sound. Good job!

Major:

- Having supplementary tables as figures is sub-optimal and, somewhat surprisingly, I couldn't find the underlying data on <http://merenlab.org/data/> either. The URL given in the manuscript is merely a placeholder (albeit with a cute kitty). This must be fixed prior to publication.

Supplementary tables were provided as Microsoft EXCEL Spreadsheets. However, we now realize that this information was not available to Dr. Bremges. We apologize for the inconvenience. We hope that this time the reviewer can access the supplementary data.

A temporary static copy of the reproducible Workflow is available at the link below.

https://www.dropbox.com/s/zjbo41kfmzc3f6w/wolbachia_reproducible_workflow.html.gz

To download and view the file, please click the download button on the right-top, and then double click the file (once to unzip, second time to open it in your browser).

The URL http://merenlab.org/data/2018_Reveillaud_et_al_Wolbachia will be made public prior to publication.

Minor:

- From the introduction (and as a non-*Wolbachia*/mosquito-expert), it is not immediately clear why *Culex pipiens* was selected as a host and why it is suitable. The authors mention *Culex pipiens* only once before, close to some *Drosophila* stuff, which was a bit confusing at first. I suggest to explain or motivate this better, 1-2 sentences should suffice.

We agree with the reviewer this information was missing. *Culex pipiens* species actually represent a key natural model for *Wolbachia* infections as almost 100% of the *C. pipiens* population is infected, and display an unusual series of canonical and bidirectional sperm-egg incompatibilities. We have now updated the introduction section to clarify this.

The authors report that "only" 50-70% of the metagenomic reads are contained in the assembly (line 121). This is nothing to worry about but, in fact, the norm. Nevertheless,

the authors should explain the "high" percentage of unmapped reads to readers not familiar with the "poor" state-of-the-art in metagenome assembly. Suitable references putting their "low" mapping rates into context are: Sczyrba et al., Nat Methods 2017 (COI: I'm a co-author of this benchmarking study); Delmont et al., Nat Microbiol 2018; ...

We thank the reviewer for pointing that out. Our estimated percentages of unmapped reads appeared as relatively low considering metagenome assembly challenges due to the complexity and diversity of microbial communities, as well as the presence of the host genome. We have now updated the manuscript to note this point with the suggested references.

- Occasionally, the language is overly complicated and could be simplified, e.g. line 62: "concomitant to" could be replaced by "due to" (which might spare non-native readers the lookup in a dictionary).

We agree with the reviewer and now have replaced the word 'concomitant' with 'simultaneous'.

I also encourage the authors to critically re-read their manuscript and break up stray long and nested sentences (e.g. line 64: "Over the last decade [...]" could easily be split up into 2-3 sentences). However, I wish to emphasize that the authors did a great job already and this last point is largely nitpicking.

We took the reviewer's suggestion, and broke up long sentences into shorter ones (e.g., line 64). We also studied the manuscript again to identify overly complicated sections and simplified our language.

Signed,
Andreas Bremges

We appreciate Dr. Bremges' time and are thankful for his insightful comments.

Reviewer #3 (Remarks to the Author):

This is a well written and state-of-the-art analysis of a very surprising finding, that of plasmids in the Wolbachia cells of individual wild captured *Culex pipiens*. I can find no major criticisms of the manuscript, but do have some minor suggestions as listed below

Minor comments:

Please review for the use of “transposasee” and “transposon”; I think sometimes the latter is what is meant, not the former.

We thank the reviewer for this clarification. We referred to IS110 as a transposase throughout the text because it was initially identified via homology to a gene encoding the transposase enzyme. However, we have re-evaluated our terminology and used TE (transposable element).

Abstract; line 44: can delete “using state of the art... strategies”.

We modified the abstract as suggested.

Line 48: commas needed after pWCP and*C. pipiens*

We thank the reviewer for their careful reading. We now have fixed this.

Line 63-64: are a current focus. Otherwise it sounds like THIS study)

We have replaced the word ‘current’ with the word ‘intense’ in our revision to clarify the meaning.

Line 108; sequenced

We have now corrected this.

Line 115: add Plasmid of.... (name)

We have modified this line accordingly in our revision.

Line 17: add shotgun ILLUMINA sequencing

We have now added Illumina into this sentence.

Line 12: do you need 'back' ?

We have removed 'back' from the sentence.

Lines 139-143: I think this is very interesting; to my knowledge this had not been done for individuals from the same population to look at SNPs (variants) in *Wolbachias*; if so it might need to be referenced.

We thank the reviewer for their attention. To the best of our knowledge, this is the first time this approach was used in this context.

Lines 147-,...: increased individual sequence ("gene") coverages could have been due to LGTs, as the Hotopp lab has shown; however your data down the line is very convincing otherwise.

We agree with the reviewer that the increase in gene coverages could be due to LGT. However, since we observed increased coverage for entire contigs, we reasoned that they represented whole genomic regions rather than individual genes laterally transferred.

Line 207: mention what antibiotics do and why you used it (its in the Methods, however)

We now have updated the Methods section to clarify that *Wolbachia* uninfected mosquitoes can be obtained by treating the host with antibiotics, typically tetracycline, which inhibits protein synthesis, or rifampicin, which interferes with nucleic acid synthesis (Kohanski et al. 2010). In addition we have also updated the main text to provide more information: "Tetracycline treatment eliminates *Wolbachia* from its hosts and is commonly used to generate uninfected lab lines".

Line 253: ...pWCP and the *Wolbachia* genome.....

We have modified the text accordingly.

Lines 314-315: a bit confusing; maybe detail this sentence a bit more

We agree with the reviewer this can be confusing and have now further detailed that the aim was to tie genomic data to environmental data.

Line 344: accessory genes?

We have removed the word “accessory” from the sentence in our revision to avoid confusion.

Line 361:from our OTHER MAGS

We have modified the sentence accordingly.

We thank the reviewer for their time, encouraging comments, and detailed input.

REVIEWERS' COMMENTS:

Reviewer #1 (Remarks to the Author):

The authors have addressed all my previous comments and they now convincingly show, with the addition of nanopore sequencing, that an extrachromosomal plasmid-like element is associated with *Wolbachia* in *Culex* mosquitoes.

My only remaining question is why the authors don't show the data from the PCRs of untreated *C. quinquesfasciatus* and treated *C. pipiens* (L252-255)? I think that showing the results would more clearly highlight the association between *Wolbachia* and the putative plasmid.

Lisa Klasson

REVIEWERS' COMMENTS:

Reviewer #1 (Remarks to the Author):

The authors have addressed all my previous comments and they now convincingly show, with the addition of nanopore sequencing, that an extrachromosomal plasmid-like element is associated with *Wolbachia* in *Culex* mosquitoes.

My only remaining question is why the authors don't show the data from the PCRs of untreated *C. quinquesfasciatus* and treated *C. pipiens* (L252-255)? I think that showing the results would more clearly highlight the association between *Wolbachia* and the putative plasmid.

Lisa Klasson

We agree with Dr. Klasson, and are thankful for her input. We have now included in our supplementary material a gel image for a PCR experiment using primers 263F and 2127R that compares an untreated *C. pipiens* sample (positive control) with an antibiotic-treated *Wolbachia*-free *C. pipiens* sample (Supplementary Figure 2).

We thank Dr. Klasson's attention to detail.